# A Case of Specificity: How Does the Acoustic Voice Quality Index Perform in Normophonic Subjects?

**Christina Batthyany** [1,*], **Youri Maryn** [1,2,3,4,5,6,*], **Ilse Trauwaen** [7], **Els Caelenberghe** [8], **Joost van Dinther** [1], **Andrzej Zarowski** [1] **and Floris Wuyts** [9]

[1] European Institute for ORL-HNS, Department of Otorhinolaryngology and Head & Neck Surgery, Sint-Augustinus GZA, Wilrijk, 2610 Antwerp, Belgium; Joost.vanDinther@gza.be (J.v.D.); Andrzej.Zarowski@gza.be (A.Z.)

[2] Department of Speech, Language and Hearing Sciences, University of Ghent, 9000 Ghent, Belgium

[3] Faculty of Education, Health and Social Work, University College Ghent, 9000 Ghent, Belgium

[4] Ecole de Logopédie, Faculty of Psychology and Educational Sciences, Université Catholique de Louvain, 1348 Louvain-la-Neuve, Belgium

[5] Faculty of Medicine and Health Sciences, University of Antwerp, 2000 Antwerp, Belgium

[6] Phonanium, 9160 Lokeren, Belgium

[7] De Spreekplank, 8400 Ostend, Belgium; ilse.trauwaen@gmail.com

[8] Logo-Land, 2930 Brasschaat, Belgium; info@babbelkous-logopedie.be

[9] Lab for equilibrium investigations and aerospace, University of Antwerp, 2000 Antwerp, Belgium; floris.wuyts@uantwerpen.be

* Correspondence: christinabatthyany@outlook.be (C.B.); youri.maryn@gza.be (Y.M.)

**Abstract:** The acoustic voice quality index (AVQI) is a multiparametric tool based on six acoustic measurements to quantify overall voice quality in an objective manner, with the smoothed version of the cepstral peak prominence (CPPS) as its main contributor. In the last decade, many studies demonstrated its robust diagnostic accuracy and high sensitivity to voice changes across voice therapy in different languages. The aim of the present study was to provide information regarding AVQI's and CPPS's performance in normophonic non-treatment-seeking subjects, since these data are still scarce; concatenated voice samples, consisting of sustained vowel phonation and continuous speech, from 123 subjects (72 females, 51 males; between 20 and 60 years old) without vocally relevant complaints were evaluated by three raters and run in AVQI v.02.06. According to this auditory-perceptual evaluation, two cohorts were set up (normophonia versus slight perceived dysphonia). First, gender effects were investigated. Secondly, between-cohort differences in AVQI and CPPS were investigated. Thirdly, with the number of judges giving G = 1 to partition three sub-levels of slight hoarseness as an independent factor, differences in AVQI and CPPS across these sub-levels were investigated; for AVQI, no significant gender effect was found, whereas, for CPPS, significant trends were observed. For both AVQI and CPPS, no significant differences were found between normophonic and slightly dysphonic subjects. For AVQI, however, this difference did approach significance; these findings emphasize the need for a normative study with a greater sample size and subsequently greater statistical power to detect possible significant effects and differences.

**Keywords:** acoustic voice quality index; smoothed cepstral peak prominence; normative data

## 1. Introduction

In the last few decades, it has been a challenge to standardize the evaluation of overall voice quality in practice. Until now, auditory-perceptual evaluation of voice quality has been considered to be the gold standard due to its accessibility and the perceptual nature of voice quality. Subsequently, it

is often used to verify the validity of other evaluation methods [1]. Several assessment tools have been developed to standardize this perceptual rating of voice quality in practice, of whom the GRBAS (grade, roughness, breathiness, asthenia, and strain) scale [2], proposed by the Japan Society of Logopedics and Phoniatrics, and the CAPE-V (consensus auditory-perceptual evaluation of voice) [3], proposed by the American Speech-Language and Hearing Association, are two well-known and widely-used scales. Nevertheless, the subjective nature and the wide range of factors related to listener, stimulus, and scale, influencing intra- and inter-judge reliability, still remains a major drawback [1,4]. Consequently, our interest in objective assessment of voice quality has grown and has been the subject of many studies.

Acoustic assessment is a much-discussed and appealing method of objective voice evaluation and is becoming increasingly used in clinical voice practice and research due to its non-invasive and accessible character and its relatively low cost [5]. To overcome the limited validity of single acoustic parameters and also recognize the multidimensionality of voice, Maryn et al. [5] developed the acoustic voice quality index (AVQI), a multiparametric tool based on six acoustic measurements, to quantify overall voice quality (an elaborate description of the acoustic measurements is represented in the section 'Materials and Methods'). This tool (Phonanium, Lokeren, Belgium) is the result of stepwise multiple linear regression analysis of thirteen acoustic measures to sort out the most robust combination yielding a single number predicting dysphonia severity, ranging from 0 to 10. The lower this number, the less dysphonia and the better the voice quality. These measures are all obtained within the software Praat (Paul Boersma and David Weenink, Institute of Phonetic Sciences, University of Amsterdam, The Netherlands). Analysis of both sustained vowels and continuous speech samples is implemented, since both sample types offer valuable information in voice quality assessment. Sustained vowels are relatively unaffected by individual speech characteristics such as speech rate, dialect, intonation, and articulation, and other factors like phonetic context and stress. On the other hand, continuous speech is more representative of daily speech and, therefore, can be considered to be more 'ecologically valid' [6]. Several studies in the last decade already demonstrated the robust diagnostic accuracy of AVQI [5,7–24], its consistent and high concurrent validity [5,7–17,19,21–25] and its high sensitivity to voice changes across voice therapy [8,12,16,18,21]. Furthermore, studies validating the use of AVQI in different languages showed diagnostic accuracy according to inter-language phonetic differences [9–14,16–25], although AVQI was originally developed for Dutch speakers.

The main contributor of AVQI is the smoothed version of the cepstral peak prominence (CPPS), which represents the distance between the first rahmonic's peak and the point with equal quefrency on the regression line through the smoothed cepstrum. The more periodic a voice signal, the more harmonic the spectrum and the higher the CPPS value. It was introduced in the field of voice assessment by Hillenbrand et al. [26] and Hillenbrand and Houde [27], and, meanwhile, many studies have proven it to be a reliable and valid measure of overall voice quality, especially of breathiness [5,8,28–36].

Subsequently, it can be concluded that AVQI, with CPPS as its main contributor, is a very promising tool to measure overall voice quality and record voice therapy outcome in practice. A current limitation of AVQI and CPPS is, however, the lack of normative data. Diagnostic thresholds to differentiate between normophonic and dysphonic voices have been proposed for mostly treatment-seeking subjects using receiver operating characteristics (ROC) curve analysis (Table 1). However, studies investigating the performance of AVQI in strictly normophonic subjects are scarce. Such normative data are valuable to contribute to the interpretation of dysphonic voice samples and thus increase the clinical utility of AVQI and CPPS. Therefore, the primary purpose of the present study was to acquire a normative data set for AVQI and its main contributor, CPPS, in Praat and to evaluate the influence of gender. The second aim was to evaluate the ability of AVQI and CPPS to distinguish the true normophonic cohort from the subclinical slight dysphonia cohort. Finally, their ability to differentiate between the several sub-levels of subclinical dysphonia was assessed.

**Table 1.** List of acoustic voice quality index (AVQI) thresholds in previous studies, investigating AVQI in different languages, with the version of AVQI used, the language, the number of evaluated voice samples, and sensitivity and specificity level and reference.

| AVQI Version | Language | Number of Voice Samples | Threshold | Sensitivity | Specificity | Reference |
|---|---|---|---|---|---|---|
| first | Dutch | 251 | 2.95 | 74% | 96% | Maryn et al. [5] |
| first | Dutch | 39 | 2.95 | 85% | 100% | Maryn et al. [8] |
| first | Dutch | 50 | 3.19 | 92% | 73% | Maryn et al. [11] |
| first | Dutch | 50 | 3.66 | 85% | 80% | Maryn et al. [11] |
| first | German | 50 | 3.05 | 98% | 75% | Maryn et al. [11] |
| first | French | 50 | 3.07 | 97% | 70% | Maryn et al. [11] |
| first | English | 50 | 3.29 | 90% | 90% | Maryn et al. [11] |
| first | English | 50 | 3.25 | 95% | 82% | Maryn et al. [11] |
| first | German | 61 | 2.70 | 79% | 92% | Barsties et al. [10] |
| first | Australian English | 107 | 3.46 | 82% | 92% | Reynolds et al. [9] |
| first | Dutch | 60 | 2.80 | 91.7% | 87.5% | Barsties et al. [15] |
| second 02.02 | Lithuanian | 264 | 3.31 | 71.7% | 88% | Barsties et al. [20] |
| second 02.02 | Korean | 1.524 | 3.33 | 90.0% | 96.5% | Kim et al. [19] |
| second 02.02 | Finnish | 50 | 2.35 | 82.1% | 95.5% | Kankare et al. [14] |
| second 02.02 | Finnish | 200 | 2.87 | 79.6% | 86.2% | Kankare et al. [22] |
| second 02.02 | Lithuanian | 264 | 3.31 | 78.1% | 92.0% | Uloza et al. [18] |
| second 02.02 | Lithuanian | 184 | 2.97 | 83.8% | 93.7% | Uloza et al. [13] |
| second 02.02 | Japanese | 336 | 3.15 | 72.5% | 95.2% | Hosokawa et al. [12] |
| second 02.02 | Dutch | 60 | 2.43 | 100% | 93.6% | Barsties et al. [15] |
| third 03.01 | Dutch | 1058 | 2.43 | 78.5% | 93.2% | Barsties et al. [7] |
| third 03.01 | Japanese | 455 | 1.41 | 84.4% | 85.6% | Hosokawa et al. [16] |
| third 03.01 | Spanish | 183 | 2.28 | 74.8% | 94.6% | Delgado et al. [17] |
| third 03.01 | German | 218 | 1.85 | 72% | 90% | Barsties et al. [21] |
| third 03.01 | French | 120 | 2.33 | 59.8% | 100% | Pommée et al. [24] |
| third 03.01 | Brazilian Portuguese | 50 | 1.10 | 57.8% | 100% | Englert et al. [37] |
| third 03.01 | Brazilian Portuguese | 53 | 1.16 | 86% | 80% | Englert et al. [23] |
| third 03.01 | Brazilian Portuguese | 53 | 1.56 | 88.6% | 100% | Englert et al. [23] |
| | **Mean** | | 2.67 | | | |

## 2. Materials and Methods

### 2.1. Participants

Normative data were collected from 123 subjects (72 females and 51 males) between 20 and 60 years old. Participants were not selected at random but were recruited by way of snowball sampling, since there was an appeal on volunteers such as family members, friends, and acquaintances. They mainly originated from the West Flemish region of Belgium, Europe. All participants included did not have any voice-related complaints, were not diagnosed with vocal pathology, and were not seeking voice treatment.

### 2.2. Voice Recordings

All subjects were asked to sustain the vowel /a:/ for at least 5 s and to read a Dutch phonetically balanced text at comfortable pitch and loudness. Both types of voice samples were recorded using an AKG C420 head-mounted condenser microphone (AKG Acoustics, München, Germany), digitized at a sampling rate of 44.1 kHz and a resolution of 16 bits using the Computerized Speech Lab model (KayPentax, Lincoln Park, NJ, USA), and saved in WAV-format. The microphone was positioned at 8–10 centimeters and at a 45° azimuthal angle from the mouth. Recording took place in an anechoic

audiometric cabin with a low mean ambient noise of 38 dBA. Vowel samples were edited to include only the medial 3 s and continuous speech samples were cropped to contain only the first two sentences of the speech task.

*2.3. Overall Dysphonia Ratings*

Using the software Praat both vowel samples and continuous speech samples were edited and emerged resulting into a concatenation of the first two sentences of the speech task, a pause of two seconds, followed by the medial three seconds of the vowel /a:/. The concatenated samples were evaluated by three raters, of whom one experienced speech-language pathologist and two final-year speech-language pathology students with minimal experience with dysphonic voices and voice pathology. Perceptual overall voice quality (i.e., hoarseness) was evaluated using the first parameter 'Grade' (i.e., G) following the GRBAS scale [2], proposed by the Japan Society of Logopedics and Phoniatrics. As recommended by Wuyts et al. [38], a 4-point equal-appearing interval scale was used (i.e., 0 = normally or absence of hoarseness, 1 = slightly hoarse, 2 = moderately hoarse, 3 = severely hoarse). Rating occurred fully separately and independently. Only those voice samples that were unanimously rated as G = 0 by all three raters, were considered to be true normophonic. Consequently, two cohorts were set up, with the first cohort consisting of 83 subjects with no perceived dysphonia and the second cohort consisting of 40 subjects with slight perceived dysphonia. Furthermore, three sub-levels of slight hoarseness were obtained according to the number of judges rating G as 1 (1 = majority normophonia, 2 = majority slight dysphonia, 3 = unanimously slight dysphonia).

*2.4. Acoustic Measures and AVQI*

Because certain acoustic measures used in AVQI are only valid for voiced segments, the continuous speech samples needed to be edited. Detection and extraction of voiced segments from the first two sentences of the speech task were performed in the software Praat using the extraction Praat-script from Maryn et al. (2009) [5], an algorithm based on three criteria, proposed by Parsa and Jamieson (2001) [6]. A segment was considered to be voiced if (a) sound energy exceeded 30% of the overall sound energy, (b) zero crossing rate was less than 1500 Hz, and (c) the normalized autocorrelation peak was above 0.3. Subsequently, the medial three seconds of the vowel /a:/ were appended to the voiced segments, resulting in a concatenation sample with a correspondent single sound waveform. The six acoustic measures implemented in AVQI were obtained using the software Praat. First, harmonics-to-noise-ratio (i.e., HNR) was specified as the base-10-logarithm of the ratio between the periodic energy and the noise energy multiplied by 10. Secondly, shimmer local (i.e., SL) was the absolute mean difference between the amplitudes of successive periods divided by the average amplitude. Thirdly, the shimmer local dB (i.e., SLdB) was obtained as the base-10-logarithm of the differences between the amplitudes of successive periods multiplied by 20. Fourthly, the general spectral slope (i.e., Slope) was calculated as the difference between the energy in the 0–1 kHz range and the energy in the 1–10 kHz range of the long-term average spectrum. Fifthly, the spectral trendline inclination (i.e., Tilt) was measured as the difference between the energy in the 0–1 kHz range and the energy in the 1–10 kHz range of the trendline through the long-term average spectrum. Finally, the sixth and main acoustic measure implemented in AVQI, smoothed cepstral peak prominence (i.e., CPPS), was calculated as the distance between the first rahmonic's peak and the point with equal quefrency on the regression line through the smoothed cepstrum. To determine the CPPS in the software Praat on a Windows or Mac computer, the following steps were completed: First, click on the "Analyze periodicity –" menu, choose "To PowerCepstrogram . . . ", and then complete the "Sound: To PowerCepstrogram" form as shown in Figure 1. Second, query the resulting PowerCepstrogram object by clicking on the "Query –" menu, choose the "Get CPPS . . . " option, and then finally complete the "PowerCepstrogram: Get CPPS" form

as shown in Figure 2. The result of these steps is that the CPPS is shown as a value in dB in a "Praat Info" screen. Ultimately, AVQI (v.02.06) was calculated according to the following regression formula:

$$\text{AVQI} = [(3.295 - (0.111 \times \text{CPPS}) - (0.073 \times \text{HNR}) - (0.213 \times \text{SL}) + (2.789 \times \text{SLdB}) - (0.032 \times \text{Slope}) + (0.077 \times \text{Tilt})) \times 2.208] + 1.797$$

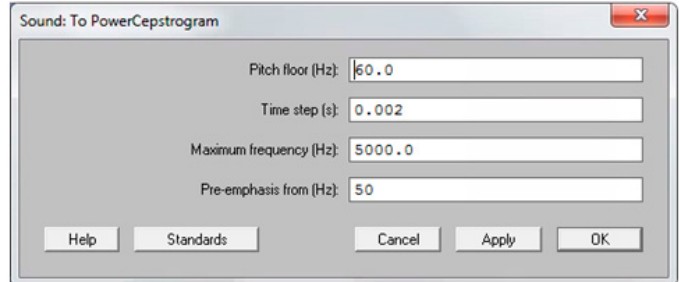

**Figure 1.** Printscreen with the arguments for the parameters to retrieve the powercepstrogram in the software Praat, for Windows computers (identical settings are used for Mac computers).

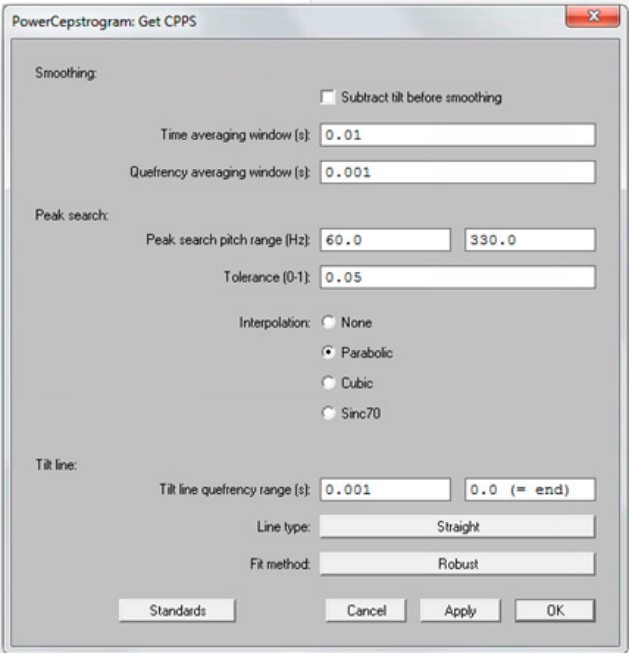

**Figure 2.** Printscreen with the arguments for the parameters to retrieve the smoothed cepstral peak prominence (CPPS) in the software Praat, for Windows computers (identical settings are used for Mac computers).

*2.5. Statistical Analysis*

Statistical analysis was completed using the software package SPSS for Windows version 15 (SPSS Inc., Chicago, IL, USA). Non-parametric statistical tests were used for small sample sizes (N < 30), as for larger sample sizes (N ≥ 30) parametric statistical tests were used. Assumptions of normality were checked and verified using the one-sample Kolmogorov–Smirnov test. First, gender effects for AVQI and CPPS were investigated using the Student t-test. Secondly, between-cohort differences in AVQI and CPPS were investigated using the Mann–Whitney U test. Thirdly, differences across the normophonic cohort and the three sub-levels of slight dysphonia were investigated using the omnibus Kruskal–Wallis H test and post-hoc Mann–Whitney U tests. All results were considered statistically

significant at p < 0.05. Upper and lower cut-off values were calculated using the standard deviation value: mean + and − (2 × standard deviation), respectively.

## 3. Results

### 3.1. Normative Data for AVQI and CPPS

Descriptive data of AVQI and CPPS in normophonic and slightly dysphonic subjects are presented in Table 2, for both men and women. In the true normophonic cohort, a mean AVQI score of 2.48 was found. The upper limit of the 95% prediction interval was 4.04. For CPPS, mean values of $CPPS_{vowel}$, $CPPS_{sentences}$, and $CPPS_{combination}$ were 15.86, 7.45, and 14.27, respectively. Lower limits of their 95% prediction intervals were 10.84, 5.91, and 10.63, respectively.

**Table 2.** Descriptive norm-referencing data of AVQI and CPPS. PI: prediction interval, SD: standard deviation.

| | Gender | Normophonic | | | | | Subclinical | | | | | All | | | | |
|---|---|---|---|---|---|---|---|---|---|---|---|---|---|---|---|---|
| | | N | Mean | SD | Lower PI | Upper PI | N | Mean | SD | Lower PI | Upper PI | N | Mean | SD | Lower PI | Upper PI |
| AVQI | F | 52 | 2.60 | 0.67 | 1.26 | 3.94 | 20 | 2.92 | 0.83 | 1.26 | 4.58 | 72 | 2.68 | 0.73 | 1.22 | 4.14 |
| | M | 31 | 2.28 | 0.91 | 0.46 | 4.10 | 20 | 2.71 | 0.90 | 0.91 | 4.51 | 51 | 2.45 | 0.92 | 0.61 | 4.29 |
| | F + M | 83 | 2.48 | 0.78 | 0.92 | 4.04 | 40 | 2.81 | 0.86 | 1.09 | 4.53 | 123 | 2.59 | 0.82 | 0.95 | 4.23 |
| CPPS vowel | F | 52 | 15.08 | 2.04 | 11.00 | 19.16 | 20 | 14.17 | 1.88 | 10.41 | 17.93 | 72 | 14.82 | 2.03 | 10.76 | 18.88 |
| | M | 31 | 17.18 | 2.71 | 11.76 | 22.61 | 20 | 16.06 | 3.25 | 9.56 | 22.56 | 51 | 16.74 | 2.96 | 10.82 | 22.66 |
| | F + M | 83 | 15.86 | 2.51 | 10.84 | 20.88 | 40 | 15.11 | 2.79 | 9.53 | 20.69 | 123 | 15.62 | 2.62 | 10.38 | 20.86 |
| CPPS sentences | F | 52 | 7.59 | 0.73 | 6.13 | 9.05 | 20 | 7.76 | 1.05 | 5.66 | 9.86 | 72 | 7.64 | 0.82 | 6.00 | 9.28 |
| | M | 31 | 7.22 | 0.79 | 5.64 | 8.80 | 20 | 7.19 | 1.10 | 4.99 | 9.39 | 51 | 7.21 | 0.91 | 5.39 | 9.03 |
| | F + M | 83 | 7.45 | 0.77 | 5.91 | 8.99 | 40 | 7.48 | 1.10 | 5.28 | 9.68 | 123 | 7.46 | 0.88 | 5.70 | 9.22 |
| CPPS combination | F | 52 | 13.80 | 1.49 | 10.82 | 16.78 | 20 | 13.23 | 1.63 | 9.97 | 16.49 | 72 | 13.64 | 1.54 | 10.56 | 16.72 |
| | M | 31 | 15.08 | 2.06 | 10.96 | 19.20 | 20 | 14.37 | 2.53 | 9.31 | 19.43 | 51 | 14.80 | 2.26 | 10.28 | 19.32 |
| | F + M | 83 | 14.27 | 1.82 | 10.63 | 17.91 | 40 | 13.80 | 2.18 | 9.44 | 18.16 | 123 | 14.12 | 1.95 | 10.22 | 18.02 |

### 3.2. Gender Effects

No significant gender effect was found for AVQI (Table 3; t = 1.673, p = 0.101). For CPPS, however, significantly higher values of $CPPS_{vowel}$ (t = −3.734, p < 0.001) and $CPPS_{combination}$ (t = −3.028, p = 0.004) were found in males and significantly higher values of $CPPS_{sentences}$ (t = 2.182, p = 0.032) in females (Table 3).

**Table 3.** Gender effect of AVQI and CPPS.

| | | 52 F vs 31 M |
|---|---|---|
| AVQI | Statistical test | Independent-samples student t test |
| | Test value | 1.673 |
| | Sign. | p = 0.101 |
| CPPS vowel | Statistical test | Independent-samples student t test |
| | Test value | −3.734 |
| | Sign | p < 0.001 |
| CPPS sentences | Statistical test | Independent-samples student t test |
| | Test value | 2.182 |
| | Sign. | p = 0.032 |
| CPPS combination | Statistical test | Independent-samples student t test |
| | Test value | −3.028 |
| | Sign | p = 0.004 |

### 3.3. Normal Versus Subclinical Voice Quality

Although mean AVQI was higher and mean $CPPS_{vowel}$ and mean $CPPS_{combination}$ were lower in the subclinical group (Table 4), none of the differences between normal and subclinical voice quality levels reached significance.

**Table 4.** Comparison of mean AVQI and CPPS values of subjects with normal versus subclinical dysphonic voice quality with corresponding significance.

|  | Normophonic | Subclinical | Statistic | Sign. |
|---|---|---|---|---|
| **AVQI** | N = 83<br>Mean = 2.48<br>SD = 0.78 | N = 40<br>Mean = 2.81<br>SD = 0.86 | Mann-Whitney U = 1303.5 | p = 0.054 |
| **CPPS vowel** | N = 83<br>Mean = 15.86<br>SD = 2.51 | N = 40<br>ean = 15.11<br>SD = 2.79 | Mann-Whitney U = 1385 | p = 0.138 |
| **CPPS sentences** | N = 83<br>Mean = 7.45<br>SD = 0.77 | N = 40<br>Mean = 7.48<br>SD = 1.10 | Mann-Whitney U = 1573 | p = 0.64 |
| **CPPS combination** | N = 83<br>Mean = 14.27<br>SD = 1.82 | N = 40<br>Mean = 13.80<br>SD = 2.18 | Mann-Whitney U = 1421.5 | p = 0.198 |

### 3.4. Subclinical Levels of Dysphonia

AVQI and CPPS data for the different subclinical levels of dysphonia are provided in Table 5. As more judges rated G as 1 (i.e., slight dysphonia), the higher mean AVQI values were observed. A significant difference was found between the true normophonic cohort and the second subclinical dysphonia cohort (Figure 3a, U = 392.00, p = 0.028). For $CPPS_{vowel}$, a declining trend was observed with a significant difference between the true normophonic and the second subclinical dysphonia cohort (Figure 3b, U = 422.00, p = 0.048). For $CPPS_{sentences}$ and $CPPS_{combination}$ no consistent trend was observed (Figure 3c,d, respectively). Comparisons with the third subclinical cohort were not conducted since this cohort included only three subjects and the statistical power consequently would be too low.

**Table 5.** Differences in AVQI and CPPS between subclinical levels of dysphonia.

| N Judges G = 1 | AVQI | CPPS Vowel | CPPS Sentences | CPPS Combination |
|---|---|---|---|---|
| **0** | N = 83<br>Mean = 2.48<br>SD = 0.78 | N = 83<br>Mean = 15.86<br>SD = 2.51 | N = 83<br>Mean = 7.45<br>SD = 0.77 | N = 83<br>Mean = 14.27<br>SD = 1.82 |
| **1** | N = 22<br>Mean = 2.57<br>SD = 0.75 | N = 22<br>Mean = 15.75<br>SD = 2.78 | N = 22<br>Mean = 7.60<br>SD = 1.16 | N = 22<br>Mean = 14.29<br>SD = 2.82 |
| **2** | N = 15<br>Mean = 3.03<br>SD = 0.93 | N = 15<br>Mean = 14.52<br>SD = 2.57 | N = 15<br>Mean = 7.32<br>SD = 0.99 | N = 15<br>Mean = 13.38<br>SD = 1.87 |
| **3** | N = 3<br>Mean = 3.56<br>SD = 0.80 | N = 3<br>Mean = 13.40<br>SD = 3.63 | N = 3<br>Mean = 7.34<br>SD = 1.50 | N = 3<br>Mean = 12.33<br>SD = 3.01 |

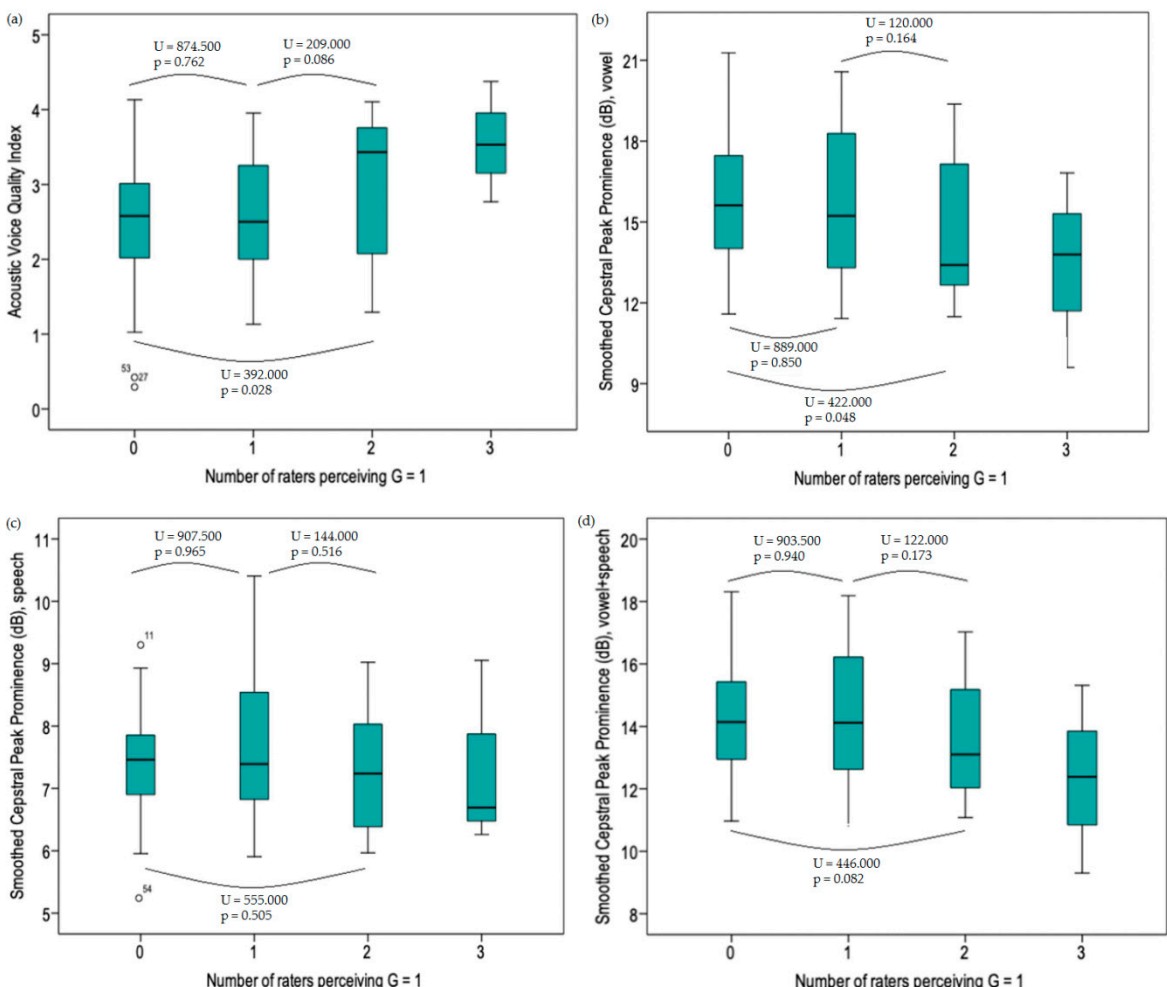

**Figure 3.** (**a**–**d**) Differences between mean AVQI and CPPS values of the normal cohort and sub-levels of subclinical slight dysphonia presented by box plots with corresponding significance.

## 4. Discussion

This study investigated the performance of AVQI and CPPS in a strictly normophonic subject group to enhance the interpretation of dysphonic voice samples and to increase the clinical utility of these measures in the assessment of overall voice quality. In 2009, Maryn et al. [5] developed the multiparametric tool AVQI and found a threshold score of 2.95 using receiver operating characteristics (ROC) curve analysis. This value needs to be interpreted as a cut-off score to discriminate between normophonic and dysphonic voices with a weighted sensitivity and specificity of 74% and 96%, respectively. In the present study, an upper cut-off value of 4.04 was found, which implies a very high specificity of 97.5% but low sensitivity. To our knowledge, only one other normative study on AVQI has been conducted. In 2017, Barsties et al. [39] performed AVQI analysis on concatenated voice samples of 123 vocally healthy Lithuanian speaking individuals and investigated the influence of age and gender. They reported a mean AVQI score of 2.32 and an upper limit value of 3.90, which approaches our findings. Similar to the present study, no significant gender effects were found. Consequently, one could suggest that AVQI is relatively unaffected by gender-based differences in vocal anatomy and physiology. In the present study, mean AVQI values were higher in subclinical subjects compared to normophonic subjects. This finding, however, did not reach significance. Furthermore, as more judges rated G as 1 (i.e., slight dysphonia), the higher mean AVQI values were observed. A significant difference was only found between the true normophonic cohort and the second subclinical dysphonia cohort.

It is also interesting to compare our CPPS values to those found in other studies with normative data, also using CPPS measurement in the software Praat. Latoszek et al. [40] performed CPPS measurements on concatenated voice samples of 530 normophonic voices and found a mean CPPS value of 11.92 (SD = 2.15). Phadke et al. [41] reported a mean value of CPPS of 13.9 (SD = 1.9) and 10.5 (SD = 1.2) for 40 normophonic vowel samples and speech samples, respectively. Heman-Ackah et al. [35] reported a mean CPPS value of 4.77 (SD = 0.97) for 87 normophonic speech samples using SpeechTool software and reported a cut-off value of 4.0 discriminating between normal and dysphonic voices with a sensitivity and specificity of 92.4% and 79% respectively, using receiver operating characteristic (ROC) curve analysis. These differences in CPPS values can be explained by different settings and methodology within and among software packages. Furthermore, comparisons with CPPS values found in the present study can only be carried out when similar settings in Praat are used (Figures 1 and 2). In the present study, specific significant gender effects were found for CPPS. Significantly higher values of $CPPS_{vowel}$ (t = −3.734, p < 0.001) and $CPPS_{combination}$ (t = −3.028, p = 0.004) were found in males and significantly higher values of $CPPS_{sentences}$ (t = 2.182, p = 0.032) in females (Table 5). It is known that the type of speech task causes important variability in the perceptual-auditory evaluation of voice quality. A previous study conducted by Maryn et al. [42] showed that the two speech tasks—i.e., sustained vowels and continuous speech—yielded significant differences in their ratings of degree of dysphonia severity (i.e., Grade). There is a tendency towards significantly higher G scores for sustained vowel samples than for continuous speech samples. The gender effect of these different types of speech tasks has however never been investigated. Based on the present study, the presumption could be made that, due to gender-related anatomical and/or physiological differences, males are somehow more fit to produce a clean normophonic sustained vowel and females are somehow more fit to produce a normophonic continuous speech task. This is, however, merely an assumption, which could be interesting to explore in future studies. Sustained vowel samples and concatenated samples show the same trend, which can be explained by the fact that in this study the sustained vowel task represents a greater contribution to the concatenated sample than the continuous speech task does.

There are some limitations regarding this study that are worth mentioning. First, the perceptual evaluation was performed by three raters, of whom two were final-year speech-language pathology students with only limited experience with dysphonic voices. Secondly, subjects mainly originated from the West-Flanders region of Belgium and were recruited by way of snowball sampling, since there was an appeal to volunteers such as family members, friends, and acquaintances. The relevance of this limitation can, however, be questioned, since multiple studies in the last decade already showed that AVQI is relatively unaffected by inter-language phonetic differences. Consequently, the influence of dialect can be negligible. Finally, only 123 subjects between 20 and 60 years old were included in the present study. In the future, a normative study with a greater number of subjects divided into different age groups is warranted to investigate whether AVQI and CPPS are able to differentiate significantly between the true normophonic cohort and the subclinical slight dysphonia cohort, as well to explore possible age effects. If AVQI and/or CPPS are found to be such sensitive measures to remark such small differences between normophonia and slight dysphonia, they could be useful as complementary tools in practice to support perceptual evaluation of voice quality in case of doubt.

## 5. Conclusions

In conclusion, the present study provided norm-referencing data of AVQI and CPPS to increase their clinical utility in practice. For AVQI, no significant gender effects were found. Consequently, one could suggest that AVQI is relatively unaffected by gender-based differences in vocal anatomy and physiology. For both AVQI and CPPS, no significant differences were found between normophonic and slightly dysphonic subjects, which emphasizes the need of a normative study with a greater sample size and subsequently greater statistical power to detect possible significant effects and differences. Future studies should be conducted with the newest version of AVQI (v.03.01).

**Author Contributions:** Conceptualization, Y.M.; methodology, Y.M., I.T., and E.C.; software, Y.M.; formal analysis, F.W.; data curation, F.W.; writing—original draft preparation, C.B.; writing—review and editing, Y.M., J.v.D, and A.Z.; supervision, J.v.D. and A.Z.

**Funding:** This research received no external funding.

**Conflicts of Interest:** Youri Maryn has financial interest in an instructional and commercial website-based company (Phonanium) that provides information and sells software described in this manuscript.

## Abbreviations

| | |
|---|---|
| AVQI | acoustic voice quality index |
| CAPE-V | consensus auditory-perceptual evaluation of voice |
| CPPS | smoothed version of the cepstral peak prominence |
| G | grade |
| GRBAS scale | grade, roughness, breathiness, asthenia and strain scale |
| HNR | harmonics-to-noise-ratio |
| SL | shimmer local |
| SLdB | shimmer local dB |
| Slope | general spectral slope |
| Tilt | spectral trendline inclination |

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
