# Peer review of "A Case of Specificity: How Does the Acoustic Voice Quality Index Perform in Normophonic Subjects?"

_applsci, doi:10.3390/app9122527_

Round 1
Reviewer 1 Report
The Acoustic Voice Quality Index (AVQI) is a multiparametric tool consisting of a weighted combination of six acoustic measurements to quantify overall voice quality in an objective manner, with Cepstral Peak Prominence (CPPS) as its main contributor. This manuscript reports some results on the AVQI’s and CPPS’s performance in normophonic non-treatment-seeking subjects.
This paper has many positive aspects. However, the abstract should conform to the template provided in the Applied Sciences journal site. Also presentation of the paper needs improvements. More details are given below.
Authors
The template of the Applied Sciences journal suggests that e-mail addresses from all the authors should be provided.
Abstract
In the template it is written ‘A single paragraph of about 200 words maximum.’ It is very alarming that the abstract is almost 400 words.
Lines 19, 27, 38, 43. The template is specific, that no headings should be included in the abstract.
In line 19, ‘acoustic Voice Quality Index’ should be ‘Acoustic Voice Quality Index’.
In line 20, the full name for the abbreviation CPPS should be provided.
Authors need to change the abstract according to the template.
Introduction
In line 54 reference numbers in brackets (and for every reference). Again, in the template it is written ‘References should be numbered in order of appearance and indicated by a numeral or numerals in square brackets’.
In lines 54 and 55, the full name of the abbreviations should be provided.
Provide a reference for line 61 (subject of many studies…).
In line 64, ‘et al’ should be ‘et al.’ Same throughout the manuscript.
In line 66 the reference should be beside the name of the authors (Maryn et al. (5))
In line 65, (based on six acoustic measurements..) The six measurements should be mentioned (or it should be mentioned that a detailed description is provided in materials and methods).
In line 70, ‘program Praat’ should be ‘software Praat’.
A very large table (table 1) with many references is included in the introduction with no discussion by the authors. A discussion about the table and its relevance should be included.
Materials and Methods
In line 116, ‘a resolution of 6 bits using…’. Do you mean 16 bits? Can you please elaborate?
In line 178, details for SPSS software should be provided.
In gender effects (lines 198-201) significant differences were found for genders concerning CPPS. The authors needs to explain this point.
Results
Tables 2 and 3 should be included in a separate page in landscape mode.
Figures 3, 4, 5, 6 should be in vector format.
In figure 3, the numbers (inside the figure) are of different size and fonts. It should be corrected.
I suggest that the font in the figures should be the same as in the manuscript (Palatino Linotype)
Page 7 seems somewhat disorganized to me. I suggest that you move all the tables of page 7 (tables 4, 5, 6, 7 ) in a separate page (without text).
Discussion
In line 257, a reference is required.
In line 242, ‘needs be interpreted’ should be ‘needs to be interpreted’.
In line 259, 260 the references should be written after the name of the authors (same throughout the manuscript).
Sentence in lines 260-264 needs to be rewritten.
In line 280, ‘sIight‘ should be ‘slight’.
References 37, 38 do not match the names of the authors. Authors should check the references throughout the manuscript.
Abbreviations
A list of abbreviations should be provided in the end.
Author contributions and funding sections should also be included as presented in the template.
Author Response
Feedback is provided in the attached file.

Reviewer 2 Report
Overall thoughts:
This manuscript presents new statistical findings and normative data on an extant functional instrument of voice. This is a fine paper that leans clinical, and therefore the fit at app. sci. should be considered by the editor. Within the manuscript, the authors clearly and transparently determine a a clinical threshold for the AVQI, a somewhat new measure of voice quality. Their finding is not in line with existing literature, but that could be because the authors are performing a first of its kind (as far as this reviewer is aware) prospective characterization of this tool for differentiating normal and subclinical populations. I have only minor revisions, listed below to improve clarity and reduce the overall number of tables and figures:
Intro:
for the uninitiated, what makes Praat "easy"?
Methods:
resolution is almost assuredly 16 bits (not 6)
spelling - "cabine" did you mean 'cabin'? Consider 'room' or 'chamber' here.
please list the manufacturer of the chamber
Figure 1 and 2 - I would suggest removing these and putting this information into a Table
Results:
Tables 4 and 5 are sufficiently described in the text - suggest to move to a digital supplement
Combine Figure 3-6 into 1 figure with 4 panels, and make font size of text within the figures larger
Author Response
Feedback is provided in the attached document.

Reviewer 3 Report
Even if the paper comes with a very unusual title, the argument it explores is deserving attention. Furthermore, all the analysis is correctly performed ad well described. I have minor revisions to suggest the authors before the paper’s publication.
Some references are too old to be worthy.
Abstract is too long and using “(1) methodology”, …, is a bad attitude.
The use of tables in the introduction should be avoided.
English writing style should be adapted to a more formal scientific writing, thus removing the use of “we” and the contracted form.
Author Response

(The authors gave the same response as above.)

Reviewer 4 Report
Topic is quite interesting. To set up normophonic AVQI threshold values is of great importance and more research is needed, in my opinion.
Some comments:
- In line 116, a resolution of 6 bits is reported. If this value is the number of bits per speech sample, this number is too low, it could be a mistake. Please check.
- In lines 126-128, subjective evaluation methodology is described with three judges, one of them is an experienced judge and two others only have minimal experience. It seems that all scores are evaluated with the same weighting, which it seems to me not the best option. The authors recognizes that this is a limitation of the study (lines 268-270), but subjective evaluation is critical in this study.
- In Figures 1 and 2, differences between Mac and Windows screenshoots are irrelevant for this study. Please consider to remove one of them (a or b).
- Please consider to remove Tables 4 and 5, as all data is presented in Table 2 and lines 198-201 in a more clear way. Tables 6 and 7 could also be avoided if proper text with statistical numbers is added in lines 205-207.
- Table 6 statistical test, with a p-value of 0.054, close to statistical significance, could be clear if substituted by a box plot graphic, showing also percentiles.
- For subclinical levels, a table with a format similar to Table 3, instead of Table 8 and Table 9 would be more clear.
- In lines 218-220, it's stated that for CPPS_{combination} the mean value in the second subclinical cohort was significantly lower ... , with a p value of 0.082. Usually, a p-value of 0.05 is used as threshold value for statistical significance.
Author Response

(The authors gave the same response as above.)
